# Toxicity of Deltamethrin to Zebrafish Gonads Revealed by Cellular Biomarkers

**Adriana Petrovici [1,†], Stefan-Adrian Strungaru [2,†], Mircea Nicoara [3],**
**Madalina Andreea Robea [3], Carmen Solcan [1] and Caterina Faggio [4,*]**

[1] Department of Molecular Biology, Histology and Embryology, Faculty of Veterinary Medicine, University of Agricultural Science and Veterinary Medicine Ion Ionescu de la Brad, 8, Mihail Sadoveanu Alley, 700489 Iasi, Romania; p.adriana6@yahoo.com (A.P.); carmensolcan@yahoo.com (C.S.)

[2] Department of Research, Faculty of Biology, Alexandru Ioan Cuza University of Iasi, Bd. Carol I, 20A, 700505 Iasi, Romania; stefan.strungaru@uaic.ro

[3] Department of Biology, Faculty of Biology, Alexandru Ioan Cuza University of Iasi, Bd. Carol I, 20A, 700505 Iasi, Romania; mirmag@uaic.ro (M.N.); madalina.robea11@gmail.com (M.A.R.)

[4] Department of Chemical, Biological, Pharmaceutical and Environmental Sciences, University of Messina, Viale Ferdinando Stagno d'Alcontres, 31-98166 S.Agata-Messina, Italy

*   Correspondence: cfaggio@unime.it; Tel.: +39-090-676-5213

†   These authors contributed equally to this work.

**Abstract:** Deltamethrin is responsible for health and reproduction problems both in mammals and aquatic organisms. In this study, zebrafish adults were exposed for 15 days to 0.25, 0.5, 1, and 2 $\mu$g L$^{-1}$ non-lethal concentrations of deltamethrin, knowing that is used worldwide on agricultural crops. We investigated the chronic effects of deltamethrin on gonads by histopathological examination, immunohistochemistry, and immunofluorescence using biomarkers for apoptosis (anti-p53, anti-H2A.XS139ph antibodies, and TUNEL assay), oxidative stress (anti-Cox4i1 antibody) and proliferation (anti-PCNA antibody). Among the histopathological changes, the apoptotic response was elevated in ovary and testis of deltamethrin exposed groups as it was seen in the IHC and IF for p53, H2A.XS139ph, and confirmed by TUNEL assay. These were observed in the case of all studied concentrations compared with the control group. Thereby, the gonadal tissue exhibited an up-regulated activity of this cell-death signaling markers, while the proliferation marker (PCNA) increased in the ovary due to its presence not only in primary growth and cortical-alveolar stage follicles but also in atretic follicles, meanwhile decreased notably in the testis. Cox4i1, a mitochondrial marker, decreased both in ovary and testis during deltamethrin treatment, probably inhibited by the overproduction of the free radicals after pesticide exposure.

**Keywords:** pyrethroids; gonads toxicity; apoptosis; immunofluorescence; biomarkers

## 1. Introduction

Pesticides are harmful not only for the organisms they were designed for, but also for other species that encounter them through water and food [1–5]. In the past decades, it was revealed that more and more water organisms develop pathologies due to water pollution with these compounds [6–9]. Synthetic pyrethroids have been for a long time considered to be one of the safer insecticides but recent studies reveal important implications in pathology. Probably even more worrying is that also human diseases are starting to be linked with pesticide usage. Recent studies incriminate the impact of pesticides on human health problems like cancer, asthma, diabetes, Parkinson's disease, leukemia, and impaired cognitive abilities in children [10]. Deltamethrin (DM) is a synthetic type II pyrethroid extensively used in pest control. The highest concentrations (59.74 $\mu$g g$^{-1}$) were detected in river

sediments in Vietnam, meanwhile, significant concentrations (55.00 µg g$^{-1}$) were also detected in adipose tissue adjacent to mammary tumors of female dogs from Brazil and in dust samples from residential areas in Berlin (54.50 µg kg$^{-1}$) [11].

Regarding the presence of deltamethrin in aquatic ecosystems, various studies concerning its negative effects on various species from the freshwater environments have been conducted in *Daphnia magna* [12,13], *Coenagrion puella* [14], *Danio rerio* [15–17], *Carassius carassius* [18], *Eriocheir sinensis* [19], *Hypophthalmichthys molitrix* [20], and *Salmo trutta fario* [21].

Research studies conducted on zebrafish model (*Danio rerio*) with various concentrations of deltamethrin revealed an inhibition of acetylcholinesterase (0.52 µg L$^{-1}$, 5.20 µg L$^{-1}$, and 10.4 µg L$^{-1}$) [22,23]; embryonic development retardation, larvae malformations, and chorion surface tension reduction (>0.1 µg L$^{-1}$) [24]; oogenesis lagging (0.5 µgL$^{-1}$ and 1 µgL$^{-1}$) [25]; swimming speed and depth alteration (0.15 µg L$^{-1}$, 1.5 µg L$^{-1}$, 3.75 µg L$^{-1}$, 7.5 µg L$^{-1}$, and 15 µg L$^{-1}$) [26]; body axis curvature, pericardial edema and developmental neurotoxicity (10 µg L$^{-1}$) [17,27]. In *Carassius carassius* DM influenced the cardiac rhythm and the force of contraction (0.1–30.0 µm) [18], meanwhile, in *Carassius auratus gibelio* determined an oxidative stress response in the liver and intestine (2 µg L$^{-1}$) [28]. In lethal concentrations it induced degeneration and necrosis in the gills and kidneys of *Cyprinus carpio* (29 µg L$^{-1}$ and 41 µg L$^{-1}$) [29] and also decreases the erythrocytes number, the hematocrit, the hemoglobin and total protein content in blood plasma (13 µg L$^{-1}$) [30].

The increasing popularity of zebrafish as a model organism is due to his many advantages in behavioral research, molecular biology, developmental biology, genetics, toxicology, drug discovery, oncology, and neuroscience research [31–33]. It is an excellent tool to study the mechanisms of pesticide toxicity and the alterations they induce in aquatic organisms. Considering the lack of information regarding the reproductive impairments associated with chronic exposure to DM in zebrafish, the study aimed to continue the research for the testis and ovaries to complete the image of the lesions produced by deltamethrin-based insecticide exposure using histopathological examination and biomarkers for apoptosis (anti-p53, anti-H2A.XS139ph antibodies, and TUNEL assay), oxidative stress (anti-Cox4i1 antibody), and cellular proliferation (anti-PCNA antibody).

## 2. Materials and Methods

### 2.1. Experimental Animals

Zebrafish adults (6–7 months old) wild-type were purchased from different breeders to obtain a genetic diversity and to simulate a wild population that can be found in freshwater aquatic environments. A total number of 200 specimens (100 males and 100 females) were housed in two aquariums (90 L each) with dechlorinate tap water for one month (accommodation period). The animals were strictly maintained and treated according to EU Commission Recommendations [34] on guidelines for the accommodation and care of animals used for experimental and other scientific purposes, Directive 2010/63/EU of the European Parliament and of the Council of 22 September 2010 on the protection of animals used for scientific purposes [35]. This experiment has been approved by the Ethical Commission from the Faculty of Veterinary Medicine, University of Agricultural Sciences and Veterinary Medicine Iasi. The water was under constant mechanical and biological filtration (pH 7.67, conductivity 552 µS cm$^{-1}$, salinity 0.26 practical salinity units, total dissolved solids 270 mg L$^{-1}$, oxidation–reduction potential +370, and water temperature −24 °C) and changed at every 24 h to avoid intoxication with harmful nitrogen compounds resulting from degradation of organic matter (e.g., ammonia and nitrates). Each treatment aquarium had a volume of 10 L and was filled up with dechlorinate tap water that was constantly aerated by an air pump and maintained at the same parameters as in the accommodation period.

### 2.2. Experimental Design

In this study, zebrafish adults were exposed for 15 days. The experiment was conducted on five study groups (a control group and four exposed groups), each group consisting of 10 males

and 10 females (20 fish per group separated based on their sexual dimorphism) that were randomly transferred from the same housing aquarium. The deltamethrin 100 g $L^{-1}$ stock solution in solvent naphtha (petroleum) light aromatic that was used in this study, it was purchased from local market with certified quality and is the active compound of a well-known insecticide, often used worldwide in pest control so that our research gets as close as possible to the real scenario. This insecticide did not include any synergist compound according to producer certification. It is a water-soluble product and was diluted for the experimental concentrations: 0.25, 0.5, 1, and 2 μg $L^{-1}$. We conducted before three experiments where it was demonstrated that at the studied concentrations there is no mortality and these were reported in various studies (between 0.04 and 24 μg $L^{-1}$ in Canadian agricultural areas; 0.73–58.8 ng $L^{-1}$ in the Ebro Delta, Spain) [36–38]. After each 24 h the water was replaced from each aquarium and the medium with deltamethrin was redone. This was repeated for 15 days of exposure time. The control group was not exposed to any circumstance to deltamethrin. The experiment was repeated once again following the same conditions and the same protocol with the zebrafish from the other housing aquarium.

*2.3. Histological Analyses, Immunohistochemistry, Immunofluorescence and Tunel Assay for Detection of Oxidative Stress, Apoptosis, and Proliferation in Gonadal Tissue*

Every fish, after euthanization with ice-cold water, had the abdominal wall sectioned from the anus to the heart and has been subjected to the usual fixation in Bouin, dehydration and embedding techniques. Sections were cut at 5 μm each with the microtome. A number of five microscope slides from each paraffin block (anterior, mid anterior, central, mid posterior and posterior region of the gonads) were stained using the standard hematoxylin-eosin protocol and examined under light microscope Olympus CX41 and Axio Observer D1 Microscope with fluorescence from Carl Zeiss, Germany. Immunohistochemistry (IHC) and immunofluorescence (IF) staining were performed using series of antibodies markers: proliferating cell nuclear antigen (PCNA) (Gene Tex, GTX124496), cellular tumor antigen p53 (Gene Tex, GTX128135), cytochrome c oxidase subunit 4 isoform 1 mitochondrial (Cox4i) (Gene Tex, GTX124479) and histone H2A.XS139ph (phospho Ser139) (Gene Tex, GTX127342). Three slides per fish (one slide for anterior, central and caudal region of the ovaries, respectively testis) were dewaxed and microwaved for 10 min at 95 °C in 10 mmol citrate acid buffer with pH6, left 20 min to cool down, then washed twice in PBS for 5 min and incubated with primary antibodies, diluted 1:100 for p53, Cox4i, PCNA and 1:200 for H2A.XS139ph, overnight at 4 °C in a humid chamber. Next day slides were washed 3 times in PBS for 5 min and incubated with the secondary antibodies (Abcam, Goat Anti-Rabbit HRP (ab205718) diluted 1:100 and Goat Anti-Rabbit Alexa Fluor 488 (ab150077) 1:1000. Sections from immunohistochemistry were developed with 3,3'-diaminobenzidine (DAB) and finally counterstained with hematoxylin.

For identification and quantification of the apoptotic cells from the ovary and testis, the Terminal deoxynucleotidyl transferase dUTP nick end labeling (TUNEL) assay was performed. It detects the apoptotic DNA fragmentation. The protocol is similar with the one for immunohistochemistry but the difference consists in a positive control by treating a section from the slide with DNA-ase for 10 min prior to adding the TUNEL mix, a negative control-treated only with the label solution and the test sections treated with the TUNEL mix (enzyme solution and label solution) according to the manufacturers indications. The rest of the protocol was similar to IHC.

*2.4. Measurements and Analysis of Histological Modifications*

Quantification for every female of control and experimental groups was done by counting and measuring different types of oocytes by Image J analysis of the images [39]. Hematoxylin-eosin photographs (n = 6 photos from each female gonad) were collected from all the fish in each group for measurements. The numerical values extracted with the software were statistically analyzed for the ovary toxicity. The Shapiro–Wilk test, for the data distribution, was firstly applied for all sets before

conducting any comparison test. The one-way ANOVA test followed by Tukey HSD test was performed to demonstrate the significant variance of the studied variables between control and exposed groups.

## 3. Results

### 3.1. Histological Analysis of DM Toxicity on Ovary

The ovary of the control group presented all the follicular development stages in a balanced proportion such that was observed clusters of primary oocytes grouped close to the germinative epithelium or scattered amongst the cortical-alveolar and vitellogenic oocytes that were also present in an important number. They were observed as well as the mature oocytes and post-ovulatory follicles in females that recently spawned (Table 1).

**Table 1.** Rate of the previtellogenic, vitellogenic, and atretic oocytes in control and experimental groups (average ± SD n = 24 images).

| | Oocytes Types Percentages | | |
|---|---|---|---|
| | **Previtellogenic (%)** | **Vitellogenic (%)** | **Atretic (%)** |
| Control | 78 ± 1.4 | 9.5 ± 1.6 | 12.5 ± 1.1 |
| 0.25 µg L$^{-1}$ | 66.8 ± 1.7 | 8.6 ± 1.07 | 24.5 ± 1.5 |
| 0.5 µg L$^{-1}$ | 55.7 ± 1 | 5.2 ± 0.96 | 39 ± 0.7 |
| 1 µg L$^{-1}$ | 52.4 ± 2.4 | 3.5 ± 1 | 44 ± 1.4 |
| 2 µg L$^{-1}$ | 46.3 ± 1.4 | 3.1 ± 1.1 | 50.5 ± 1.2 |
| one-way ANOVA | | | |
| F statistics | 1401.7 | 150.5 | 3876.3 |
| variance | 3804.3 | 205.8 | 5698.2 |
| *p* value | <0.001 | <0.001 | <0.001 |
| Tukey HSD test without significant results | Control vs. 0.25 µg L$^{-1}$ $p = 0.065$ 1 µg L$^{-1}$ vs. 2 µg L$^{-1}$ $p = 0.759$ | | |

The group exposed to 0.25 µg L$^{-1}$ DM presented minor ovarian modifications, but we could highlight an increase in atretic oocytes, 24.53% compared to 12.5% founded in control group, a slightly decreased number of the previtellogenic and vitellogenic oocytes (Table 1), cytoplasmic fragmentation, moderate folding of the follicular membrane, presence of different stage oocytes with degenerative alterations (Table 2).

**Table 2.** Ovarian histopathological findings after deltamethrin exposure.

| | Mature Oocytes | Vitellogenin Oocytes | Pre-Vitellogenic Oocytes | Atretic Follicles | Lipofuscin | Follicular Cells Hypertrophy | Membrane Invagination | Extravascular Proteic Fluid |
|---|---|---|---|---|---|---|---|---|
| Control | + | + | + | ± | ± | − | − | − |
| 0.25 µg L$^{-1}$ | ± | ++ | ++ | + | + | + | + | + |
| 0.5 µg L$^{-1}$ | ++ | ± | + | ++ | ++ | + | + | + |
| 1 µg L$^{-1}$ | +++ | ± | ++ | ++ | ++ | ++ | ± | + |
| 2 µg L$^{-1}$ | +++ | ± | ++ | +++ | +++ | ++ | + | ± |

Results represent semi-quantitative observations (visual scores): not present (−); low presence (±), moderate presence (+), high presence (++), very high presence (+++).

After the administration of 0.5 µg L$^{-1}$ DM, the alterations accentuated presenting a score of 55.75% previtellogenic oocytes, 5.25% vitellogenic oocytes, notably low percentages compared to 78%, respectively 9.5% of the control group and very high number of atretic oocytes (39.08%). We also found an important detachment of the follicular cells layer from the zona radiata, membrane folding, vacuolization, and fragmentation of the cytoplasm of the mature oocyte (Table 2).

The group exposed to 1 µg L$^{-1}$ DM had notable ovarian degenerations and alterations. Thereby, the oogenesis was considerably reduced and the oocytes viability was affected. The detachments between zona radiata and follicular cells layer were increasing and the follicular membrane break begins to produce. The ooplasm was intensely granular or with an amorphous appearance in vitellogenic follicles, whilst the nucleus in atretic follicles suffers degeneration, the nucleoli lose their homogeneous structure and their regulate appearance, the disintegration of yolk bodies in vitellogenic oocytes was remarkably increasing, the shape of immature and primary oocytes is altered and the apoptotic process begins.

The group subjected to the highest concentrations of insecticide, 2 µg L$^{-1}$ DM, showed the most marked ovarian changes. Thus, the prevalent type of oocytes was the atretic one (50.58% ± 1.2%) where the follicles degenerated with cytoplasmic fragmentation, granulation and vacuolization, granular and theca cells debris, or even follicular rupture with the ovarian outgrowth of vitellogenic granules as an amorphous and highly acidophilic material, nuclear degeneration, and ovarian fibrosis. The intermediate follicular stages of cortical–alveolar and vitellogenic oocytes are almost lacking in the ovarian mass, mature oocytes are numerically outmoded by atretic ones, the oogonia are found in very large clusters and many of them degenerate, while the rate of previtellogenic and vitellogenic oocytes decreases considerably (46.36% ± 1.4% respectively 3.16% ± 1.1%) compared to control and the other experimental groups.

The atretic oocytes, observed at all stages of follicular development, were found more frequently in primary and vitellogenic oocytes. The primary ones were marked by the gradual degeneration of the nucleus with pyknosis and the destruction of the nuclear envelope, the deformation of the oocytes, and finally the appearance of the apoptotic bodies with an amorphous structure. Vitellogenic follicles in the atretic process showed marked hypertrophy and hyperplasia of the granulosa cells, gradual degeneration of the zona radiata with progressive loss of transversal striations, disappearance of the basal membrane between granulosa and theca cells, in injured granulosa cells, the formation of apoptotic bodies and the aggregation of chromatin in a semilunar structure, as well as the progressive resorption of vitellus by the granulosa cells transformed in large phagocytic cells.

Lipofuscin, a yellow-brown pigment-related to the aging and senescence of the cells, has been correlated also with the temporal growth arrest of the cells. It has been found that it is formed mainly in lysosomes by the peroxidation of unsaturated fatty acids in complex with proteins. Its accumulation in the ovaries of the females exposed to the toxic was dose-dependent. Thus, if in the control group, lipofuscin deposits were almost absent, in the 0.25 µg L$^{-1}$ DM showed a moderate presence, meanwhile, in the 0.5 and 1 µg L$^{-1}$ DM groups were seen in a high degree in the ovarian tissue. The highest expression of the lipofuscin was noted in the most exposed group (2 µg L$^{-1}$ DM) determining a high accumulation of complexes of degraded, peroxidized lipids, and proteins correlated with the high amount of atretic oocytes.

### 3.2. Proliferation Index of the Cells in Ovaries

By using the anti-PCNA antibody, proteins that reveal the replication of the genetic material and the cellular multiplication, the atretic follicles are easily delimited from the viable ones due to the hyperplasia of the theca cells layer in the atretic follicles and the positive marking of their nuclei in a considerably higher number than for follicular cells of viable oocytes.

A strong PCNA-positivity was observed (Figures 1 and 2) at the level of the nucleus of the oogonia that undergo the stages of oogenesis, as well as of the cortical alveolus oocytes. Considering the first observation, it was noticed that in the groups exposed to DM, the positive staining of the nucleus of the oogonia was gradually reduced with the toxic dose due to the diminution of oogenesis. Considering the diminishing toxic effects of the on the oogenesis, the replication of the DNA in the oogonia undergoes a progressive decrease, directly proportional to the doses of the compound. The PCNA-positivity in the cytoplasm of the cortical-alveolus and vitellogenic oocytes was probably correlated with mitochondrial DNA synthesis in large quantities in the mitochondria, which leads to the accumulation of an increased number of organelles in the cytoplasm of the oocyte to participate in the synthesis of the cortical alveoli.

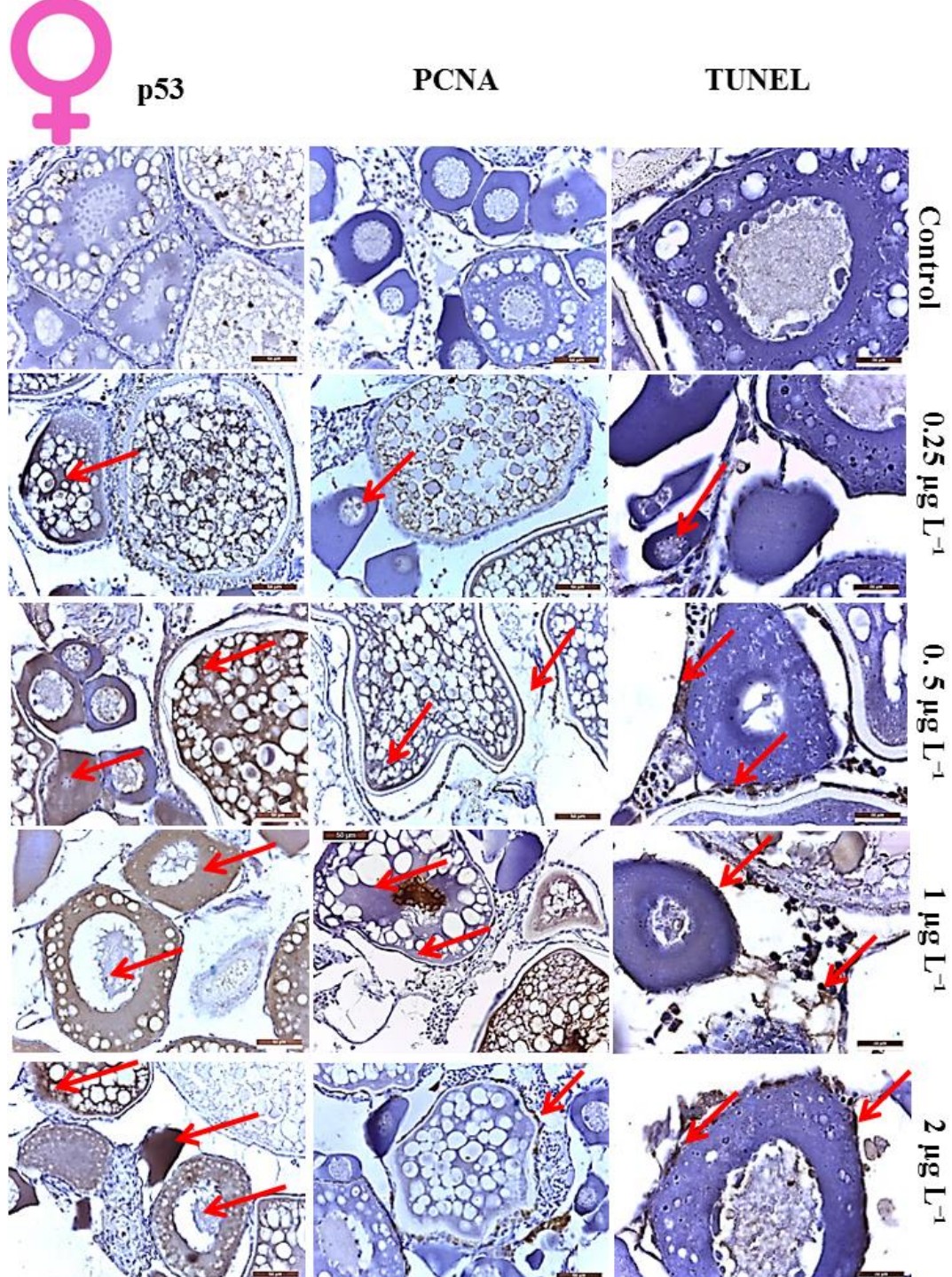

**Figure 1.** Histological analysis and immunocytochemistry of the zebrafish exposed to deltamethrin (DM) for the female gonad. Cells marked with red arrows represent examples for the positive activity of the studied markers (scale from images represents 20 μm).

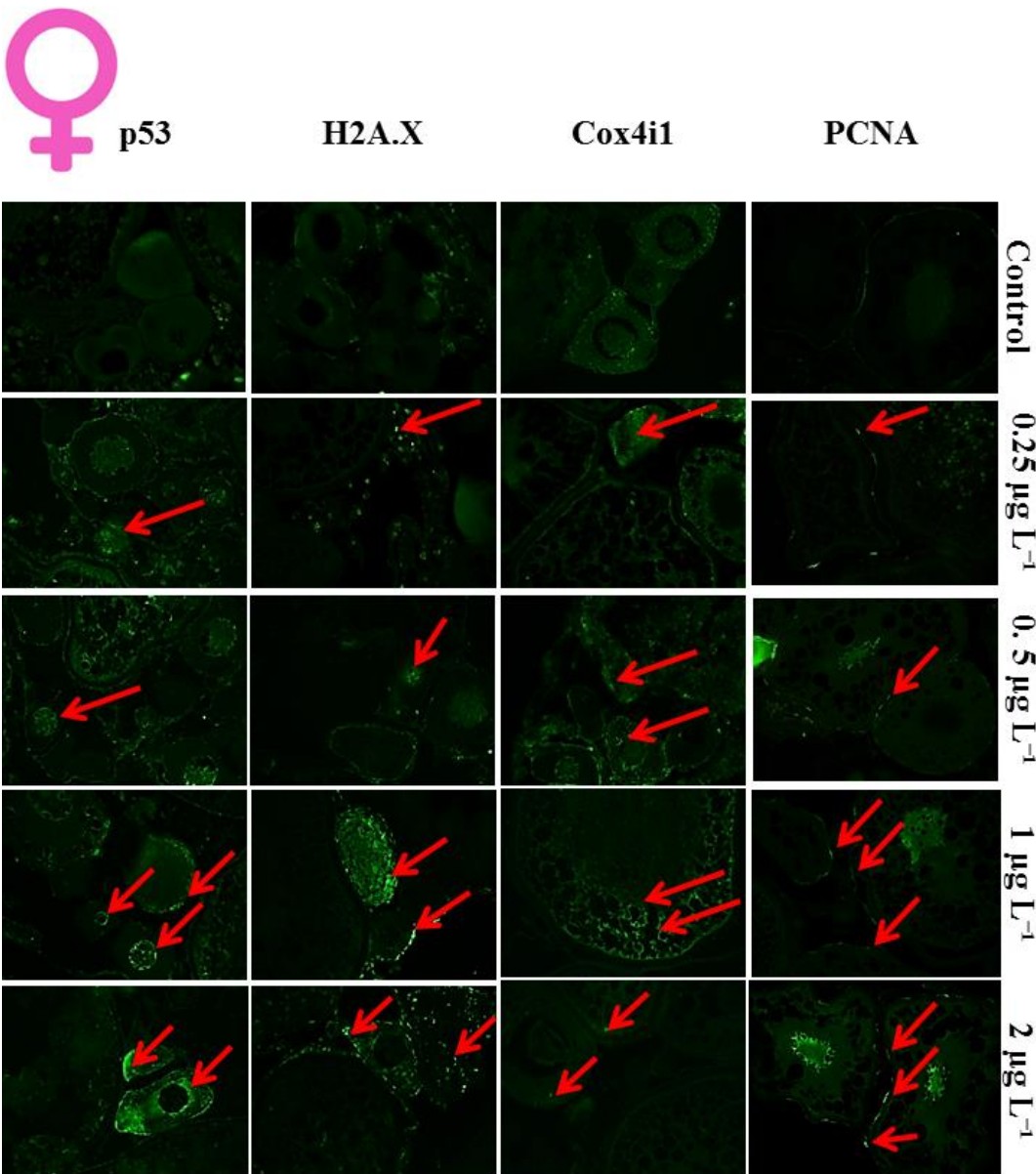

**Figure 2.** Histological analysis and immunofluorescence of the zebrafish exposed to DM for female gonad. Cells marked with red arrows represent examples for the positive activity of the studied markers.

### 3.3. Apoptotic Index of the Cells in Ovaries

After the chronic exposure of zebrafish females at DM, the ovaries presented a large number of apoptotic cells. TUNEL assay revealed an enlarged percentage of apoptotic oocytes and follicular cells in exposed groups expressed as DNA double-strand breaks present in their nucleus and evidenced with terminal deoxynucleotidyl-transferase dUTP nick end labeling (TUNEL).

The major tumor suppressor protein p53 accumulates in cells (Figures 1 and 2) in response to the appearance of DNA alterations, the activation of oncogenes or other stressors. The role at both nuclear and cytoplasmic levels of p53 has long been accepted. At the Embryology level, it behaves as a transcriptional factor that activates genes involved in apoptosis, cell cycle regulation, metabolic changes, and other processes, at cytoplasmic level initiates apoptosis by stimulating the accumulation of $Ca^{2+}$ ions in the endoplasmic reticulum (ER) and inhibits autophagy. Thus, oncosupressor proteins are considered to be preferentially located in mitochondria associated with ER membranes. As a result of increasing doses of the toxin, a gradual increase in both nuclear and cytoplasmic staining for p53 was noted. The most prominent cytoplasmic positivity was observed in the cortical alveolus and vitellogenic oocytes, most likely due to their abundance in mitochondria, ER and Golgi apparatus involved in the synthesis of cortical alveoli.

Phosphorylation of histone H2A.X at Ser-139, a sensitive marker for the presence of DNA double-strand breaks, was also quantified in the ovarian tissue of the control and exposed groups revealing an over-expression directly proportional with the concentration of the toxin administered in fish (Figure 2). These results were correlated with the other markers of the apoptosis, the p53 protein expression and the TUNEL assay for the DNA double-strand breaks revealing an increase in their expression positively associated with the quantity of DM provided.

A Cox4i deficiency was observed at the gonad level. The immunofluorescence staining with the anti-Cox4i antibody revealed a progressive reduction of the cytoplasmic presence of the mitochondrial marker (Figure 2). The diminishing of this ATP producing protein could be correlated with an increment of DNA instability as it was found by Douiev et al. [40]. These results can be correlated with other apoptotic markers used in our study.

### 3.4. Histological Analysis of Toxicity on Testis

In the control group, the seminiferous tubules showed all stages of spermatogonial development in a balanced proportion, from spermatogonia, spermatocytes, spermatids to mature spermatozoa filling the lumen of seminiferous tubules and efferent ducts (Figure 3). The group exposed to 0.25 µg $L^{-1}$ DM showed a slight but visible diminution of the spermatocytes count and size, while spermatogonia showed a minor increase in number. The third group exposed to 0.5 µg $L^{-1}$ had an intermediate evolution between in gonad toxicity that was observed in group II and the one from groups IV and V. Thus, group exposed to 1 µg $L^{-1}$ DM had a significant involution of the cysts and the number of spermatocytes and spermatids, with numerous spermatogonia clusters that were present among them. Mature spermatozoa were found in the lumen of seminiferous tubules, but in a lower percentage compared to the control group. The exposure to the highest dose of 2 µg $L^{-1}$ DM had the most notable changes. The seminiferous tubules have been significantly diminished by reducing the size and number of sperm cells. The lumen of seminiferous tubules showed only a small number of spermatozoa. However, the efferent ducts were full of mature sperm. induces, an infiltration with an acidophilic fluid has been observed at the edge of the testis of one of the subjects.

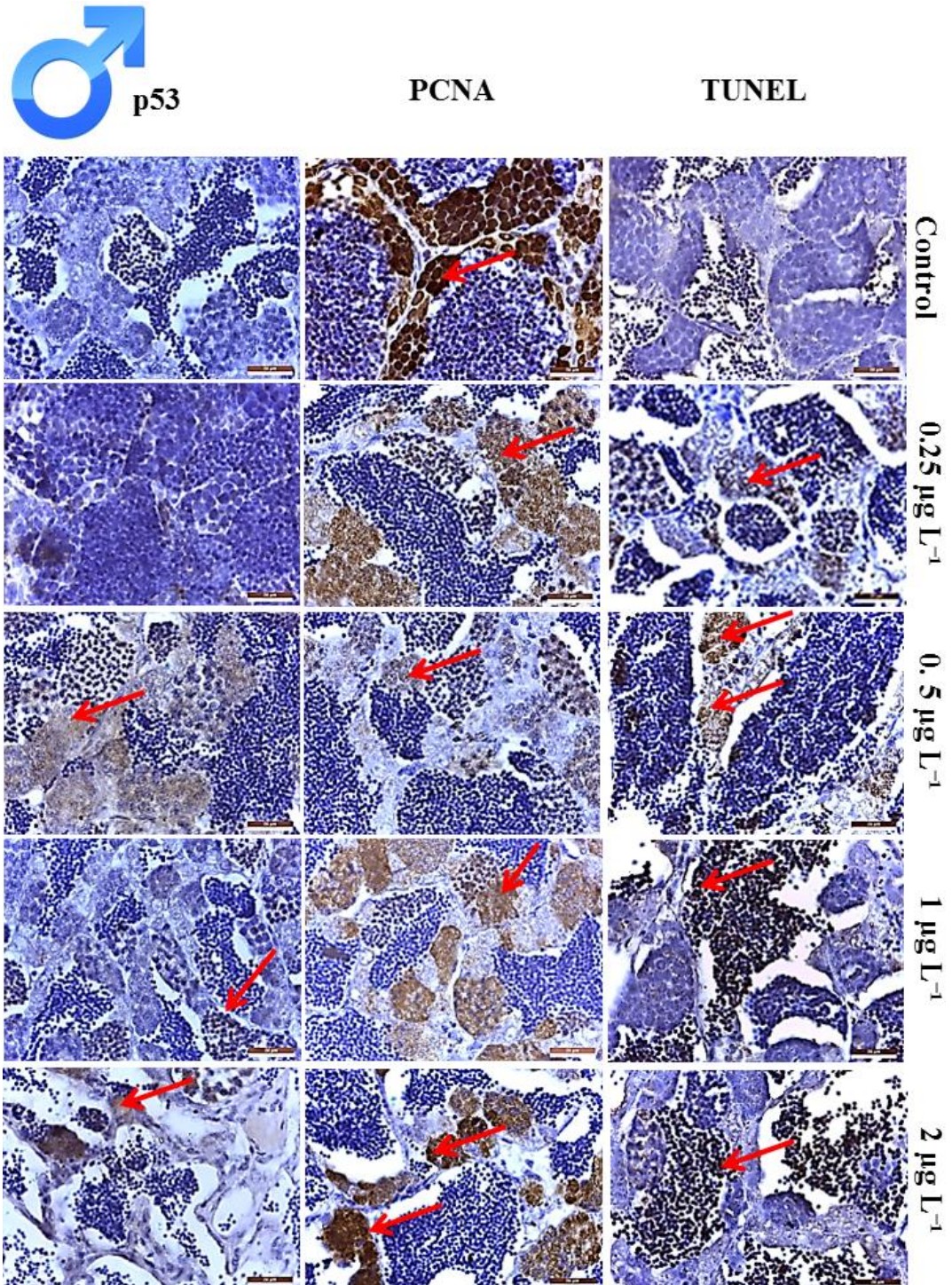

**Figure 3.** Histological analysis and immunocytochemistry of the zebrafish exposed to DM for male gonad. Cells marked with red arrows represent examples for the positive activity of the studied markers (scale from images represents 20 μm).

*3.5. Proliferation Index of the Cells in Testis*

In the testis of an adult zebrafish male, the proliferation was normally encountered in differentiated spermatogonia, but also in both Sertoli and Leydig cells. Therefore, the positive staining for PCNA markers was normally encountered.

In the present study (Figures 3 and 4), immunohistochemical staining with anti-PCNA antibody revealed the inhibition of spermatogonial proliferation by lower labeling of this component compared to the control group. The other stages were markedly positive, with the exception of mature spermatozoa that was not marked. Immunostaining with PCNA has made it easier to notice the reduction in sperm cell diameter and cellularity due to the strong positive marking that contrasts with the complete absence of positivity in mature cells. A directly proportional reduction of PCNA staining was observed with the increase of toxic exposure. It was noticed that DM has a deleterious effect on spermatogenesis.

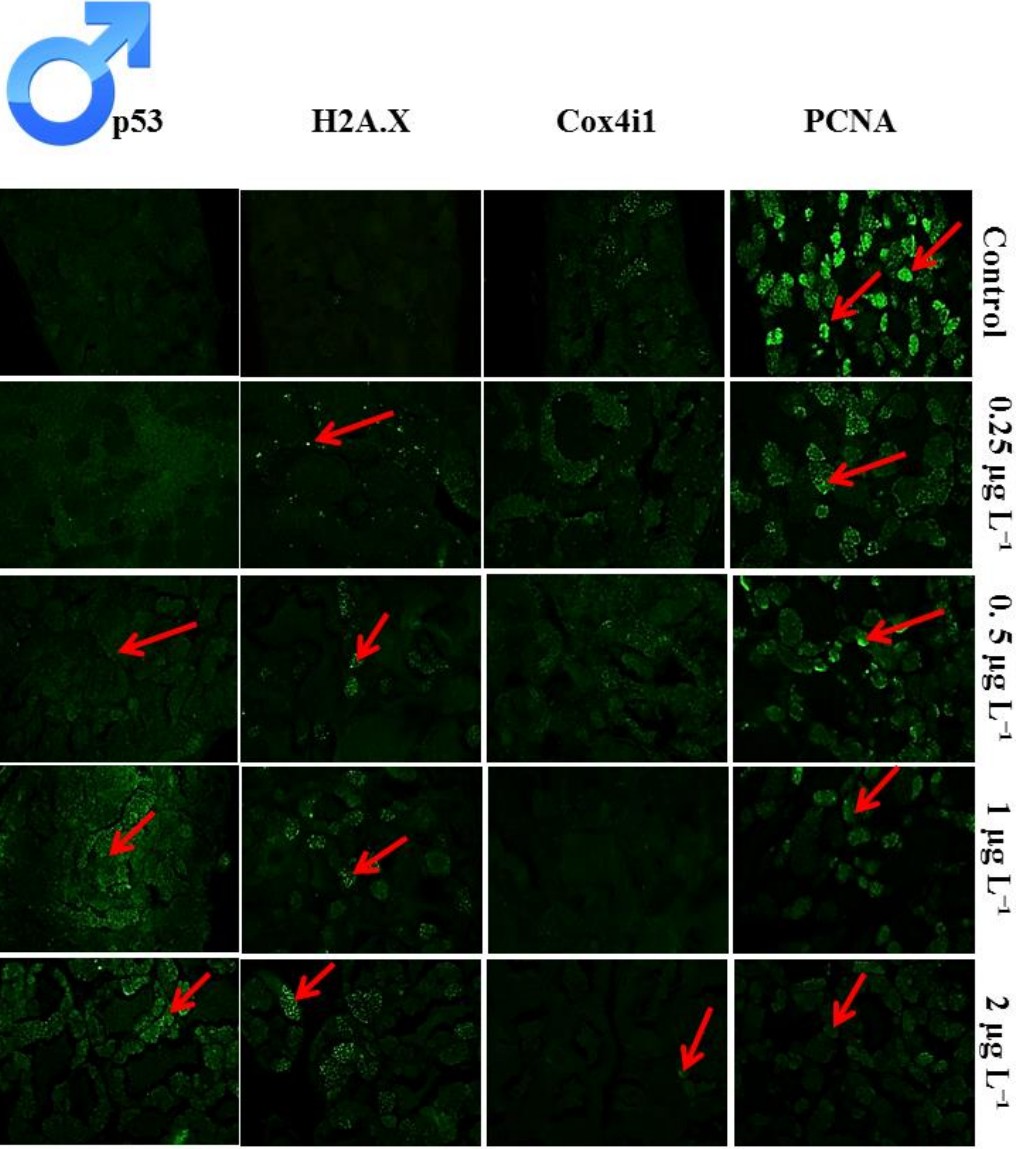

**Figure 4.** Histological analysis and immunofluorescence of the zebrafish exposed to DM for male gonad. Cells marked with red arrows represent examples for the positive activity of the studied markers.

*3.6. Apoptotic Index of the Cells in Testis*

The apoptosis in a normal process is present from the spermatogonial phase (around 20%), to spermiogenesis (10%) and meiosis (10%). However, after exposure to different doses of DM, these parameters suffered modifications. TUNEL staining detected and quantified programmed cell death at the single-cell level, based on the labeling of DNA strand breaks. Breaking of genomic DNA during

apoptosis may produce double-stranded as well as single-cell breaks ("nicks"), which can be identified by labeling free 3′-OH terminal with modified nucleotides in an enzymatic reaction.

Immunohistochemical staining for the p53 protein demonstrated a gradual increase in positive staining directly proportional to the toxic dose (Figures 3 and 4); the group exposed to the highest dose had the highest positivity of spermatogonic and spermatocytic lines. The activation of the p53 protein through DNA damage or other stresses can induce cell cycle arrest or apoptosis if the impairment cannot be repaired. By exposing zebrafish to DM, the spermatocytes suffered the major deterioration and showed the most marked positivity for the anti-p53 antibody, but also the other germinal cell stages showed an increase in p53 activity.

The quantification of the histone H2AX phosphorylated on serine 139 in control and exposed groups uphold the anterior findings. The DNA damage response hallmark, H2A.XS139ph had an increased expression in the groups that received higher doses of toxin. As shown by others, apoptotic markers used in our study, the most impaired stage was the spermatocyte one. Hence, the histone H2AX modification supports the apoptotic effects of DM on the testicular tissue. At the testicular level, a reduction of Cox4i activity was highlighted. The groups that received different doses of DM exhibited an evident diminution of Cox4i immunohistochemical staining, suggesting the nuclear DNA instability in male gonadal cells. These findings emphasize the apoptotic activity up-regulation in the testis of fish that received DM.

## 4. Discussion

The scientific literature regarding the applications of studied biomarkers for deltamethrin toxicity in zebrafish is not rich in papers focused on this subject. In general, the exposure to different doses of deltamethrin is responsible for fertility decreasing in zebrafish based on histological results of the gonads. Thereby, in female gonads, we observed an important change in the ovarian distribution of the different types of oocytes that led, in the most exposed group, to the prevalence of the atretic follicles, notable reduction of the intermediate stages and the abundance in pre-nucleolar oocytes due to decreased oogenesis. The male reproductive system suffered alterations after chronic exposure to DM. We noted fibrosis and a reduction of testes volume together with an important and progressive diminution of the spermatocysts containing spermatocytes and spermatids of the exposed groups. Several studies conducted with zebrafish confirmed our results. An exposure to 2 µg L$^{-1}$ and 5 µg L$^{-1}$ of deltamethrin for 6 days determined the reduction in egg production and histopathological changes like degenerated oocytes, degenerated follicles, disappearance of nucleolar details, and lack of nuclei in the lining of epithelial cells in ovary and degenerated epithelium with degenerated spermatids and spermatogonia, disappearance of seminiferous tubular membranes, and vacuolar changes in the epithelial lining cells in testis according to Kadiri and Gundala [41]. Similar results were concluded by Koç et al. [25] regarding the ovarian histopathological effects of acute exposure to deltamethrin in zebrafish showing that this insecticide has significant negative effects on the oogenesis. The study revealed an increase in atretic oocytes and oocytes that were depraved and grew unhealthily in exposed groups. The number of oocytes in the ovaries of the experimental groups was reduced compared to that of the control group where these changes were not observed.

Regarding the biomarkers used, they highlighted a series of aspects. For instance, proliferating cell nuclear antigen (PCNA) is normally expressed in the nuclei of mitotically active cells in the follicle epithelium of zebrafish oocytes. Korfsmeier [42] described the presence of PCNA in normal zebrafish ovary. Positively stained cells were detected in the follicle epithelium of all stages of growing oocytes and degenerating oocytes, in the nuclei of oogonia and oocytes in meiotic prophase. Follicle epithelium showed an increasing number of PCNA-positive cells during egg cell development with a very low index of <0.5% at the beginning and a maximum of >7% in stage IV oocytes. In this phase, follicle epithelium often showed groups of stained cells in limited zones of proliferation. In degenerating oocytes, over 60% of the former follicle cells were positive for PCNA. Nuclei of oogonia were always positive and the nucleus of stage III oocytes showed strong positivity. In the past years proliferating cell

nuclear antigen (PCNA) has been used to identify proliferating spermatogonia in drug discovery and toxicity studies on rat testes. D'Andrea et al. [43,44] reported a simultaneous PCNA and TUNEL assay for the determination of testicular toxicity and concluded that detection of apoptosis may be more sensitive than proliferation. Similar applications of PCNA were described in our study. Regarding this biomarker's activity, we evidenced the decreased oogenesis and spermatogenesis as the expression in mitotically and meiotically active germ cells decreased. It was revealed an increment of positive follicular cells in atretic oocytes. In the case of the p53 protein in the zebrafish model, Storer and Zon [45] described numerous aspects: the functions of cytoplasmic p53 protein are the control of apoptosis by association with mitochondria and when the normal levels of p53 are not exceeded, this protein inhibits autophagy; cytoplasmic p53 stimulates the accumulation of $Ca^{2+}$ in RE by interacting with ATP-ase and this increases the efficiency of $Ca^{2+}$ transport between RE and mitochondria, directing premalignant cells exposed to oncogenic or chemotherapeutic stress upon apoptosis. It is well accepted that many stimuli such as chemical substances and radiations can increase the p53 expression of the cells, therefore leading to apoptosis. Accordingly, in our study we prove that deltamethrin exposure of adult zebrafish determined an enhancement of the p53 expression at the gonadal level, pointing out a negative effect over the reproductive function.

It was proved by González-Rojo et al. [46] that exposure to high doses of bisphenol A in zebrafish males determines decrease in spermatocytes, an increase in apoptosis determined by TUNEL assay (BPA at 2000 μg $L^{-1}$ caused chromatin fragmentation, promoting direct reproductive effects, incompatible with embryo development), a downregulation of ccnb1 and sycp3, all of which lead to an alteration of spermatogenesis and to meiotic arrest and an upregulation of gper1 and esrrga receptors. Additionally, the highest dose promoted DNA hypermethylation and an increase in histone acetyltransferase activity, which led to the hyperacetylation of histones. Jäämaa et al. [47] used γH2AX as a rapid marker for DNA damage response and p53 as a more universal marker for cellular stress and also cleaved caspase-3, a known apoptotic marker that confirmed γH2AX suitable as an apoptotic marker. In accordance with them, we selected H2A.X.ser139 as a marker for DNA damage, together with p53 for apoptosis. Our results regarding these biomarkers evidenced that deltamethrin induces apoptosis in the zebrafish gonads, extending the previous studies regarding its negative effects on reproduction. The TUNEL assay was used to confirm the apoptotic pathway of gonadal toxic response by an evident intensification of double-stranded DNA breaks presence in the nuclei of ovarian and spermatogonial cell lines of the deltamethrin treated fish.

There were no differences found in the expression of p53 mRNA in zebrafish larvae exposed to deltamethrin by Awoyemi et al. [17] but Maalej et al. [48] observed a significant increase, showing that deltamethrin treatment-induced oxidative damage and inflammation through increasing cox-2 expression and initiates a series of cell death signaling events, including p53 activation, leading to DNA fragmentation and apoptosis.

Regarding the pathomechanism of cytochrome c oxidase deficiency, it was shown that fibroblasts derived from patients with isolated COX4-1 deficiency presented increased double-stranded DNA breaks [40]. We also observed in our study that the gonadal tissue of zebrafish exposed to different quantities of deltamethrin exhibited a decrease in the COX4i by the reduction of the intensity of the immunohistochemical signal. This was similar to TUNEL assay results and p53 and H2A.X positivity, we concluded that deltamethrin induced DNA damage and apoptosis in germ cells with the impairment of oogenesis and spermatogenesis in the case of zebrafish.

## 5. Conclusions

The cellular biomarkers used in this study evaluated the gonadotoxicity effects of deltamethrin in zebrafish adults both for males and females and may have other applications for studying the harmful effects of other chemical compounds harmful to the freshwater fish community. Both methods of investigation immunocytochemistry and immunofluorescence may be applied to bring more evidence to protect aquatic life forms against harmful pollutants released in the environment.

**Author Contributions:** Conceptualization S.-A.S., A.P., C.S. and C.F.; methodology M.N., S.-A.S. and C.S.; investigation, S.-A.S., A.P., M.A.R; data analysis S.-A.S., A.P., M.A.R.; writing—original draft preparation, S.-A.S., A.P., C.S., C.F.; writing—review and editing, C.F., S.-A.S., A.P.; supervision, M.N., S.-A.S. and C.S.; All authors have read and agreed to the published version of the manuscript.

**Funding:** S.-A.S. and M.A.R. contribution was founded by the grant "Developing innovation capacity and increasing the impact of excellence research at UAIC" funded by the Romanian Ministry of Research and Innovation within Program 1—Development of the national RD system, Subprogram 1.2—Institutional Performance—RDI excellence funding projects, Contract no. 34PFE/19.10.2018 and M.N. contribution was founded by the grant of the Romanian Ministry of Research and Innovation, CCCDI-UEFISCDI, project number 26PCCDI/01.03.2018, "Integrated and sustainable processes for environmental clean-up, wastewater reuse and waste valorization"(SUSTENVPRO), within PNCDI III.

**Acknowledgments:** We thank for the reviewers contribution that significantly improved the quality and form of this manuscript.

**Conflicts of Interest:** All authors declared that the present study did not involve any conflicts of interest.

**Ethical Approval:** This experiment has been approved by the Ethical Commission from the Faculty of Veterinary Medicine, University of Agricultural Sciences and Veterinary Medicine Iasi, Romania.

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
