# Peer review of "Toxicity of Deltamethrin to Zebrafish Gonads Revealed by Cellular Biomarkers"

_jmse, doi:10.3390/jmse8020073_

Round 1
Reviewer 1 Report
General comments.
The topic of the study is timely and interesting. The effects of the presence of Deltamethrin are worth of study indeed these authors have already assessed these effects on the nervous system of zebrafish. However, and as the authors state in previous studies, "the evaluation of toxic effects of a single contaminant does not offer a realistic estimate of the impact against aquatic ecosystems". In some instances the claims about the effects oh human health should be toned down (see lines 35-36 at the end of the abstract) and focus it on the effects on fish health/reproduction. The discussion is very poor and does not deepen on the meaning of all the results obtained in the study leaving part of them without any explanation.
The study needs corrections of English language and some style improvements.
Specific comments.
The duration of the exposures (days of treatment) should be stated at least in Materials and Methods
Were all the antibodies used at the same dilution? Dilutions should be stated for each of them.
Somnewhere in materials and methods it should be clearly stated from the 5 histologicxalo slides taken from each fish gonads how many were used for each antibody or histology or tunnel assay...It becomes impossible to know in the way it is now written Regarding histological preparations it would be interesting to know how many slides were studied of each fish/treatment to know how representative are the results. The authors say that 5 slides from each fish were examined. How representative of the whole gonad were these slides. Moreover, how representative of each histological slice were the results showed in the photomicrographs. Basically, the results were qualitative but not quantitative and thus it is no possible to extract a definate conclusion from this study. How similar are the results in each fish from the same treatment?
Table 1 needs to be completed with the results of the statistical analyses
Table 2. The meaning of the values for lipofuschin should be explained in results and discussion. What is the meaning of this marker?
Figure 2. Regarding PCNA panel, it seems that image of 0.25u/l and 1ug/l are the same but with different brightness. This is not acceptable.
I do not find PCNA immunofluorescence results in females to be clear in signal (Fig 2). A similar remark can be made for the immunocytochemistry results (Fig 1) for the same marker (PCNA). Can the authors increase the quality of the images?
The results for males are more clear than tose for females (Fig 3 and Fig 4)
Line 239. What do authors mean by "Involution of the cysts"?
The explanation of the results from the histopathological alterations at the gonad level is quite profuse, however it is difficult to infer all that from the images presented. Moreover, the presence of all these alterations in all the fish is not supported by statistical analyses and therefore it becomes impossible to make a generalization of the effect. The authors should provide data on the number of fish with these alterations and also the percentage of alterations found per group.
There is a clear lack of discusion of the results reported and in general the conclusions are not supported by the results.
Line 291 in the Discusion. Authors state that "Exposure to progressive doses of DM determined a decreased fecundity in zebrafish". In this paper a direct sffect on fecundity in zebrafish has not been studied but just the effect on some markers for the gonadal status of some fish that were treated with Deltamethrin. In order to make an statement of such an importance as it is the effect on fecundity, other parameters shoud have been measured such as the sperm volume produced or the number of eggs spawned.
The discussion is mainly focussed on the histopathological alterations found in the gonads and leaves aside the markers used for IHC an IF, in some cases as in Cox4i, H2A.XS139ph or caspase-3, completely leaving them without any discussion. Moreover, the concluding paragraph with the conclusions is just a mere repetition of the resuslts.
Author Response
Distinguished Editorial Border of Journal of Marine Science and Engineering,
We would like to thank for the effort of three reviewers that helped us to increase the quality of our manuscript with the ID jmse-662496. The corrections in the main manuscript were done in the Word document marked with yellow color.
Reviewer # 1 specific comments:
In some instances the claims about the effects on human health should be toned down (see lines 35-36 at the end of the abstract) and focus it on the effects on fish health/reproduction.
Answer: This sentence was removed from abstract
The duration of the exposures (days of treatment) should be stated at least in Materials and Methods
Answer: It was presented the exposure time in abstract. We included in materials and methods.
In this study, zebrafish adults were exposed for 15 days. The experiment was conducted on five study groups (a control group and four exposed groups), each group consisting of 10 males and 10 females (20 fish per group separated based on their sexual dimorphism) that were randomly transfer from the same housing aquarium. The deltamethrin 100 g L-1 stock solution in solvent naphtha (petroleum) light aromatic was used in this study that was purchased from local market with certified quality and is the active compound of a well-known insecticide, often used worldwide in pest control so that our research gets as close as possible to the real scenario. This is a water soluble product and was diluted for the experimental concentrations: 0.25, 0.5, 1 and 2 μg L-1. We conducted before three experiments where was demonstrated that at the studied concentrations there is no mortality and these were reported in various studies (between 0.04 and 24 μg L−1 in Canadian agricultural areas; 0.73–58.8 ng L−1 in the Ebro Delta, Spain) [36-38]. After each 24h the water was replaced from each aquariums and the medium with deltamethrin was redone. The control group was not exposed to any circumstance to deltamethrin. The experiment was repeated once again following same conditions and the same protocol with the zebrafish from the other housing aquarium.
Were all the antibodies used at the same dilution? Dilutions should be stated for each of them.
Answer: The dilution for p53, Cox4i, PCNA was 1:100 and for H2A.XS139ph 1:200.
Somewhere in materials and methods it should be clearly stated from the 5 histological slides taken from each fish gonads how many were used for each antibody or histology or tunnel assay. It becomes impossible to know in the way it is now written Regarding histological preparations it would be interesting to know how many slides were studied of each fish/treatment to know how representative are the results. The authors say that 5 slides from each fish were examined. How representative of the whole gonad were these slides. Moreover, how representative of each histological slice were the results showed in the photomicrographs. Basically, the results were qualitative but not quantitative and thus it is no possible to extract a last conclusion from this study. How similar are the results in each fish from the same treatment?
Answer: For the H&E staining, a number of 5 slides per fish were stained, representing the anterior, mid anterior, central, mid posterior and posterior region of the gonads. Concerning the immunohistochemistry, immunofluorescence and TUNEL assay, for each technique and each antibody, we stained 3 slides per fish, representative for the anterior, central and posterior region of the gonads. Therefore, for every control and treated fish, we evaluated the modifications of the gonadal tissue at the histological level by examining the aspects of the gonads at different levels (anterior, mid anterior, central, mid posterior and posterior region). Concerning the immunohistochemistry and immunofluorescence, we evaluated for every fish 3 slides per antibody representing the anterior, central and posterior region of the gonads. In this way, the modifications of the proteins expression from the whole gonad could be noted.
Table 1 needs to be completed with the results of the statistical analyses
Answer: We presented in Table 1 the results for one-way ANOVA and Tukey HSD test without significant results
Oocytes types percentages |
|||
previtellogenic (%) |
vitellogenic (%) |
atretic (%) |
|
Control |
78±1.4 |
9.5±1.6 |
12.5±1.1 |
0.25 μg L-1 |
66.8±1.7 |
8.6±1.07 |
24.5±1.5 |
0.5 μg L-1 |
55.7±1 |
5.2±0.96 |
39±0.7 |
1 μg L-1 |
52.4±2.4 |
3.5±1 |
44±1.4 |
2 μg L-1 |
46.3±1.4 |
3.1±1.1 |
50.5±1.2 |
one-way ANOVA |
|
|
|
F statistics |
1401.7 |
150.5 |
3876.3 |
variance |
3804.3 |
205.8 |
5698.2 |
P value |
***<0.001 |
***<0.001 |
***<0.001 |
Tukey HSD test |
|
Control vs 0.25 μg L-1 |
|
|
|
||
|
1 μg L-1 vs 2 μg L-1 |
|
|
|
|
Table 2. The meaning of the values for lipofuschin should be explained in results and discussion. What is the meaning of this marker?
Answer: In this table, the distinction between control and exposed groups was made by degree of impairment concerning the main histopatological lessions observed. Lipofuscin is a pigment that deposits in tissues due to an increased degradation of the cells by forming complexes of degraded, peroxidated lipids and proteins. Accumulation of lipofuscin in tissues was correlated with senescence and ageing of cells, but also with temporal arrest of cell growth, in our case, due to a progressive exposure to the toxic.
Figure 2. Regarding PCNA panel, it seems that image of 0.25u/l and 1ug/l are the same but with different brightness. This is not acceptable.
Answer: Thank you for your suggestion. We corrected the figure and according to Revewer#2 we try to make it clearer and sharper for interpretation. Please see it in the manuscript.
I do not find PCNA immunofluorescence results in females to be clear in signal (Fig 2). A similar remark can be made for the immunocytochemistry results (Fig 1) for the same marker (PCNA). Can the authors increase the quality of the images?
Answer: Thank you! We try to do our best, for quality of these images. We hope they are better for the readers and suggestive.
The exposure to different doses of Deltamethrin induced in the ovaries of the treated females an increment of the atretic oocytes. By using the PCNA marker, the atretic oocytes distinguish by the higher number of PCNA positive thecal cells, as already stated in literature by Korfsmeier, 2002. Thereby, the degenerating oocytes showed over 60% PCNA-positive cells at the follicle epithelium level. We changed some of the images to increase the clarity of the results.
Korfsmeier, K.-H. (2002). PCNA in the ovary of zebrafish (Brachydanio rerio, Ham.- Buch.). Acta Histochemica, 104(1), 73–76. doi:10.1078/0065-1281-00632
The results for males are more clear than tose for females (Fig 3 and Fig 4)
Answer: We hope they are better for the readers and suggestive
Line 239. What do authors mean by "Involution of the cysts"?
Answer: By involution of the cysts in the testis of the group exposed to 1 μg L-1 DM I was referring to the reduction of the dimensions of all the types of cysts that fill the seminiferous tubules, due to a decrease of the cells number and also of their dimensions after the toxic exposure. As we stated in the results, a decrease of the testicular volume was noticed after the treatment with DM.
The explanation of the results from the histopathological alterations at the gonad level is quite profuse, however it is difficult to infer all that from the images presented. Moreover, the presence of all these alterations in all the fish is not supported by statistical analyses and therefore it becomes impossible to make a generalization of the effect. The authors should provide data on the number of fish with these alterations and also the percentage of alterations found per group.
Answer: The amount of images needed for a complete presentation of the histopatological alterations in both ovaries and testis of control and exposed groups is considerable, this is why we chose to use proliferation (PCNA) and apoptotic markers like p53, H2A and TUNEL assay to highlight the degeneration and alterations induced by Deltamethrin in zebrafish gonads.
Nevertheless, an evaluation of the female gonad modifications after DM treatment was made using ImageJ processing programme determining the changes in number and diameter of different types of oocytes present in ovaries. The results were statistically processed using the one-way ANOVA method and Tukey test for variability and multiple comparisons.
There is a clear lack of discusion of the results reported and in general the conclusions are not supported by the results.
Answer: There were
Line 291 in the Discusion. Authors state that "Exposure to progressive doses of DM determined a decreased fecundity in zebrafish". In this paper a direct sffect on fecundity in zebrafish has not been studied but just the effect on some markers for the gonadal status of some fish that were treated with Deltamethrin. In order to make an statement of such an importance as it is the effect on fecundity, other parameters shoud have been measured such as the sperm volume produced or the number of eggs spawned.
Answer: Thank you for this correction. It his true we did a mistake in writing this sentence, this term was exhaustive used. We corrected. Exposure to different doses of deltamethrin determined a decreased fertility in zebrafish based on histological results of the gonads
The discussion is mainly focussed on the histopathological alterations found in the gonads and leaves aside the markers used for IHC an IF, in some cases as in Cox4i, H2A.XS139ph or caspase-3, completely leaving them without any discussion. Moreover, the concluding paragraph with the conclusions is just a mere repetition of the resuslts.
Answer: The discussions were according to the published results of literature. There were some results a little bit difficult to discuss because not all markers were studied in this way to compare with it. Caspase-3 has not been analysed in this paper. We changed the conclusions to avoid repetition of the results according to your observation.
Conclusions
The cellular biomarkers used in this study evaluate the gonadotoxicity effects of deltamethrin in zebrafish adults both for males and females and may have other applications for studying the harmful effects of other chemical compounds harmful to freshwater fish community. Both methods of investigation immunocytochemistry and immunofluorescence may be applied to bring more evidence in order to protect aquatic life forms against harmful pollutants released in environment.
Reviewer 2 Report
This paper report on a useful experiment in which zebrafish were exposed to deltamethrin in water at sublethal concentrations. Serious effects are seen in sections of gonadal tissue using immunohistochemical markers. It looks like the first adverse effects are already seen at 0.25 µg/L, which illustrates the high toxicity of deltamethrin to fish. Since toxicity of deltamethrin to fish has been reported in numerous studies and a well-supported benchmark value of 0.9 -3.5 µg/L, depending on the fish species, is generally accepted the novel aspect of the paper is not as a toxicity study but lies in the use of histochemical biomarkers.
However, the paper suffers from a number of quite serious insufficiencies which must be resolved before it can be reconsidered. This refers to:
The duration of exposure. The number of replicates used in oocyte counts, etc. Statistical tests showing which is the lowest exposure with an adverse effect that is significantly different from the control. This is essential. Specification of the deltamethrin formulation, including any adjuvants. Removal of excessive abbreviations in the text.There are also many errors in wording, the use of articles etc. Some corrections follow below:
Editorial remarks and suggestions
Line 1: Toxicity is “to” and effects are “on” so the “on” in line 4 is not appropriate.
Line 1: The word “chronic” is not appropriate as the exposure was for 2x24 hours (?).
Line 3: base on = based on, but better is “revealed by”
Line 4 delete “activity”
Line 1-4: Maybe the title could read: ”Toxicity of deltamethrin to zebrafish gonads revealed by cellular biomarkers”.
Line 20: Do not abbreviate deltamethrin to DM (throughout the paper), as this is quite an uncommon and also unnecessary abbreviation.
Line 20: responsible in = responsible for
Line 23: on gonads = on gonadal tissue
Line 24: do not abbreviate immunofluorescence and immunohistochemistry; just spell out these words or choose another formulation.
Line 26 – 28. The sentence “The chronic exposure of … reduction of reproductive capacity” is non-informative and can be deleted.
Line 29: indicate at what exposure levels these changes were seen, or what was the lowest exposure level where the responses were significantly different from the controls.
Line 42: preoccupying = worrying
Line 55-63: indicate the approximate exposure levels at which these effects were observed (it is no wonder that a pesticide is toxic at some dose; the point is, is it toxic at doses that are observed in the environment as a consequence of agricultural practice).
Line 68: gonadal = reproductive
Line 76: higher – higher than what? Closer – compared to what?
Line 84: salinity – in what units is this expressed (promille?)
Line 84 – 85: spell out TDS and ORP
Line 95: indicate what was the formulation (code name of product). What adjuvants were in the formulation? Any detergents or synergists? (Sometimes pyrethroid pesticide formulations are supplemented with piperonylbutoxide, an inhibitor of biotransformation, but a toxin in itself).
Line 97: presumedly, the concentrations 0.25 – 2 µg/L refer to the active ingredient?
Line 99: “and were measured in environmental samples” – What range of concentrations were measured in the cited papers? The 96h-LC50 of deltamethrin to fish is reported as 0.9 – 3.5 µg/L so the concentrations used here are already quite close to toxic levels.
Line 101: Do I understand the exposure was for 24 hours? Or were the fish that had been exposed for 24 hours exposed for another 24 hours in a new aquarium? If not, were the two groups of fish pooled in the data analysis, or were they considered as replicates?
Line 112: spell out the names of the proteins at least once, then use the abbreviation after that (e.g. “proliferating cell nuclear antigen (PCNA)”).
Line 114: Table 1: In some cases data are rounded to whole numbers, in other cases percentages are given with two decimals. Please use a harmonized presentation. It is recommended to round off to just one decimal, since count data are rarely extensive enough to support the second decimal. This also holds for the text (line 150 and further).
Line 140: Indicate what was the number of replicates from which the standard deviations were derived.
Line 142-143: Effects are suggested to be observable at the lowest exposure of 0.25 µg/L. The issue is: are these effects significant (i.e. the percentage significantly higher than in the control?)
Figure 1, Figure 2, Figure 3 and Figure 4: please adapt the ugly very large fonts for p53, PCNA and TUNEL to a font comparable with the labels on the right side of the graph.
Line 291: progressive = different (the dosing was not progressive; it was constant (or decreasing) in each aquarium.
Line 299: Is the period of six days from the paper by Kadiri & Gundala [41]? In that case “The exposure…” in line 298 should read “An exposure…”.
Line 309: astects = aspects
Line 342: Again the word “chronic” is not appropriate, unless the chronic nature of the exposure is specified.
Author Response
Reviewer # 2 specific comments:
Since toxicity of deltamethrin to fish has been reported in numerous studies and a well-supported benchmark value of 0.9 -3.5 µg/L, depending on the fish species, is generally accepted the novel aspect of the paper is not as a toxicity study but lies in the use of histochemical biomarkers.
Answer: Yes! This was the central main application in this paper since it is designated for Special Issue Biomarkers of Stress Response in Aquatic Life. Thank you!
The duration of exposure. The number of replicates used in oocyte counts, etc. Statistical tests showing which is the lowest exposure with an adverse effect that is significantly different from the control. This is essential. Specification of the deltamethrin formulation, including any adjuvants. Removal of excessive abbreviations in the text. Line 1: The word “chronic” is not appropriate as the exposure was for 2x24 hours (?)
Answer: We changed in manuscript but we keep abbreviations to avoid exhaustive repetition since we write in same manner other papers.
In this study, zebrafish adults were exposed for 15 days. The experiment was conducted on five study groups (a control group and four exposed groups), each group consisting of 10 males and 10 females (20 fish per group separated based on their sexual dimorphism) that were randomly transfer from the same housing aquarium. The deltamethrin 100 g L-1 stock solution in solvent naphtha (petroleum) light aromatic was used in this study that was purchased from local market with certified quality and is the active compound of a well-known insecticide, often used worldwide in pest control so that our research gets as close as possible to the real scenario. This is a water soluble product and was diluted for the experimental concentrations: 0.25, 0.5, 1 and 2 μg L-1. We conducted before three experiments where was demonstrated that at the studied concentrations there is no mortality and these were reported in various studies (between 0.04 and 24 μg L−1 in Canadian agricultural areas; 0.73–58.8 ng L−1 in the Ebro Delta, Spain) [36-38]. After each 24h the water was replaced from each aquariums and the medium with deltamethrin was redone. The control group was not exposed to any circumstance to deltamethrin. The experiment was repeated once again following same conditions and the same protocol with the zebrafish from the other housing aquarium.
Line 1: Toxicity is “to” and effects are “on” so the “on” in line 4 is not appropriate. Line 3: base on = based on, but better is “revealed by” Line 4 delete “activity” Line 1-4: Maybe the title could read: ”Toxicity of deltamethrin to zebrafish gonads revealed by cellular biomarkers”.
Answer: Thank you very much! This sounds better.
Toxicity of Deltamethrin to Zebrafish Gonads Revealed by Cellular Biomarkers
Line 20: Do not abbreviate deltamethrin to DM (throughout the paper), as this is quite an uncommon and also unnecessary abbreviation. Line 24: do not abbreviate immunofluorescence and immunohistochemistry; just spell out these words or choose another formulation.
Answer: We try to use this terminology throughout the paper without abbreviations but is quite annoying. We change it.
Line 20: responsible in = responsible for Line 23: on gonads = on gonadal tissue Line 42: preoccupying = worrying
Answer: Thank you! These corrections were done.
Line 26 – 28. The sentence “The chronic exposure of … reduction of reproductive capacity” is non-informative and can be deleted.
Answer: This has been deleted.
Line 29: indicate at what exposure levels these changes were seen, or what was the lowest exposure level where the responses were significantly different from the controls.
Answer: Thank you! We did changes in the manuscript. These were observed in case of all studied concentrations compared with the control group.
Line 55-63: indicate the approximate exposure levels at which these effects were observed (it is no wonder that a pesticide is toxic at some dose; the point is, is it toxic at doses that are observed in the environment as a consequence of agricultural practice).
Answer:
Line 76: higher – higher than what? Closer – compared to what?
Answer: We did changes in the manuscript
Adult (6-7 months old) zebrafish wild-type were purchased from different breeders in order to obtain a genetic diversity and to simulate a wild population that can be found in fresh water aquatic environments.
Line 84: salinity – in what units is this expressed (promille?) Line 84 – 85: spell out TDS and ORP
Answer: This was added in the manuscript.
salinity 0.26 practical salinity units, total dissolved solids 270 mg L-1, oxidation-reduction potential
Line 95: indicate what was the formulation (code name of product). What adjuvants were in the formulation? Any detergents or synergists? (Sometimes pyrethroid pesticide formulations are supplemented with piperonylbutoxide, an inhibitor of biotransformation, but a toxin in itself). Line 99: “and were measured in environmental samples” – What range of concentrations were measured in the cited papers? The 96h-LC50 of deltamethrin to fish is reported as 0.9 – 3.5 µg/L so the concentrations used here are already quite close to toxic levels.
Answer: The deltamethrin 100 g L-1 stock solution in solvent naphtha (petroleum) light aromatic that was used in this study, it was purchased from local market with certified quality and is the active compound of a well-known insecticide, often used worldwide in pest control so that our research gets as close as possible to the real scenario. This is a water soluble product and was diluted for the experimental concentrations: 0.25, 0.5, 1 and 2 μg L-1. We conducted before three experiments where was demonstrated that at the studied concentrations there is no mortality and these were reported in various studies (between 0.04 and 24 μg L−1 in Canadian agricultural areas; 0.73–58.8 ng L−1 in the Ebro Delta, Spain) [36-38]. After each 24h the water was replaced from each aquariums and the medium with deltamethrin was redone.
Line 101: Do I understand the exposure was for 24 hours? Or were the fish that had been exposed for 24 hours exposed for another 24 hours in a new aquarium? If not, were the two groups of fish pooled in the data analysis, or were they considered as replicates?
Answer: We did corrections to be more clear.
After each 24h the water was replaced from each aquariums and the medium with deltamethrin was redone. This was repeated for 15 days exposure time.
Line 112: spell out the names of the proteins at least once, then use the abbreviation after that (e.g. “proliferating cell nuclear antigen (PCNA)”).
Answer: This was corrected.
Line 140: Indicate what was the number of replicates from which the standard deviations were derived.
Answer: We ad d this information.
Table 1. Rate of the previtellogenic, vitellogenic and atretic oocytes in control and experimental groups (average±SD n=24 images )
Oocytes types percentages |
|||
previtellogenic (%) |
vitellogenic (%) |
atretic (%) |
|
Control |
78±1.4 |
9.5±1.6 |
12.5±1.1 |
0.25 μg L-1 |
66.8±1.7 |
8.6±1.07 |
24.5±1.5 |
0.5 μg L-1 |
55.7±1 |
5.2±0.96 |
39±0.7 |
1 μg L-1 |
52.4±2.4 |
3.5±1 |
44±1.4 |
2 μg L-1 |
46.3±1.4 |
3.1±1.1 |
50.5±1.2 |
one-way ANOVA |
|
|
|
F statistics |
1401.7 |
150.5 |
3876.3 |
variance |
3804.3 |
205.8 |
5698.2 |
P value |
***<0.001 |
***<0.001 |
***<0.001 |
Tukey HSD test |
|
Control vs 0.25 μg L-1 |
|
|
|
||
|
1 μg L-1 vs 2 μg L-1 |
|
|
|
|
Line 142-143: Effects are suggested to be observable at the lowest exposure of 0.25 µg/L. The issue is: are these effects significant (i.e. the percentage significantly higher than in the control?)
Answer: The statistical tests results were added in the manuscript table.
Figure 1, Figure 2, Figure 3 and Figure 4: please adapt the ugly very large fonts for p53, PCNA and TUNEL to a font comparable with the labels on the right side of the graph.
Answer: We did these corrections.
Line 291: progressive = different (the dosing was not progressive; it was constant (or decreasing) in each aquarium.
Answer: We did these corrections.
Line 299: Is the period of six days from the paper by Kadiri & Gundala [41]? In that case “The exposure…” in line 298 should read “An exposure…”.
Answer: We did these corrections.
Line 309: astects = aspects
Answer: We did these corrections.
Line 342: Again the word “chronic” is not appropriate, unless the chronic nature of the exposure is specified.
Answer: We hope now is more clearer.
Reviewer 3 Report
The use of pesticides is on the one hand a great help in achieving high plant productivity while on the other hand it is a significant threat to the living organisms. Hence the research presented like in this paper on determining hazards and pesticides are extremely important.
The title manuscript describes the threats of Deltamethrin on Zebrafish gonads. The work is very carefully done. The results are reliable and thoroughly discussed. The manuscript is well organized and clearly written.
Author Response
Reviewer # 3 specific comments:
The title manuscript describes the threats of Deltamethrin on Zebrafish gonads. The work is very carefully done. The results are reliable and thoroughly discussed. The manuscript is well organized and clearly written. Answer: Thanks a lotRound 2
Reviewer 1 Report
In the new revised version of the manuscript most of my concerns have been taken into consideration and I think the paper has improved notably. However, I still think the discussion could be improved since at its present from is not much of a discussion but a repetition of the results. The authors may argue, as they do, that there's not so much literature or experiments on that topic but I really miss some more discussion about their own findings. In any case, the experimental design is correct and the endpoint and markers studied are also relevant.
Author Response
Distinguished Editorial Border of Journal of Marine Science and Engineering,
We would like to thank for the effort of three reviewers that helped us to increase the quality of our manuscript with the ID jmse-662496. The corrections in the main manuscript were done in the Word document marked with green colour.
Reviewer # 1 specific comments:
In the new revised version of the manuscript most of my concerns have been taken into consideration and I think the paper has improved notably. However, I still think the discussion could be improved since at its present from is not much of a discussion but a repetition of the results. The authors may argue, as they do, that there's not so much literature or experiments on that topic but I really miss some more discussion about their own findings. In any case, the experimental design is correct and the endpoint and markers studied are also relevant.Answer: We try to make it clearer for the readers. Please see the changes with green colour. We did a statement regarding the lack of literature in this field for having a better discussion. This was all we could get from our study because we were focused only for gonadal aspect of these biomarkers.
Reviewer # 2 specific comments:
The authors have addressed my comments seriously and improved the paper significantly. The only issue is that the formulation of deltamethrin is not specified ("it was bought on the local market"). The point is, some deltamethrin formations are known to include a synergist, piperonyl butoxide, which is a potent inducer of radical oxygen species (due its inhibition of cytochrome P450). In many scientific papers, effects ascribed to deltamethrin (which acts upon the nervous system), could actually be due to the synergist (generating ROS and causing the type of cytoxic effects described in this paper). To avoid such contamination of the literature, it is essential that toxicologists work with pure active ingredients, or in the case of formulated products, specify the formulation (which has to list any synergists, if there are). So I would still welcome a statement by the authors that the product used by them in this paper does not contain a synergist. For the future, it is advised to test pure deltamethrin and compare its toxicity with the same exposure from a formulated product.Answer: We understand your concerns about using this product that may arise questions about its toxicity. We already published two papers with it in important Journals: Toxicity and chronic effects of deltamethrin exposure on zebrafish (Danio rerio) as a reference model for freshwater fish community (https://doi.org/10.1016/j.ecoenv.2019.01.057) ; Antagonistic effects in zebrafish (Danio rerio) behavior and oxidative stress induced by toxic metals and deltamethrin acute exposure (https://doi.org/10.1016/j.scitotenv.2019.134299).
We choose to use an insecticide that is worldwide spread and the active compound is deltamethrin. In a real case it is more likely to have a contamination with one of this product. The name is DECIS 100 EC produced by Bayer with the concentration 100 g/L deltamethrin (https://www.bvl.bund.de/SharedDocs/Downloads/04_Pflanzenschutzmittel/01_zulassungsberichte/007418-00-01.pdf?__blob=publicationFile&v=3). As we do not have any conflict of interest with this company or with their product, we did not mentioned the brand name in the manuscript. We preferred to use this insecticide, rather than a standard DM solution (for analytical purposes) because we wished to be closer to reality. Still the toxicology is an unpredictable field research. There are too many variables and interactions in environment that can influence the toxicity and effects.
We did a statement in our manuscript: The deltamethrin 100 g L-1 stock solution in solvent naphtha (petroleum) light aromatic that was used in this study, it was purchased from local market with certified quality and is the active compound of a well-known insecticide, often used worldwide in pest control so that our research gets as close as possible to the real scenario. This insecticide did not include any synergist compound according to producer certification. It is a water soluble product and was diluted for the experimental concentrations: 0.25, 0.5, 1 and 2 μg L-1.
Thank you very much for your constructive suggestions!
Reviewer 2 Report
The authors have addressed my comments seriously and improved the paper significantly. The only issue is that the formulation of deltamethrin is not specified ("it was bought on the local market"). The point is, some deltamethrin formations are known to include a synergist, piperonyl butoxide, which is a potent inducer of radical oxygen species (due its inhibition of cytochrome P450). In many scientific papers, effects ascribed to deltamethrin (which acts upon the nervous system), could actually be due to the synergist (generating ROS and causing the type of cytoxic effects described in this paper). To avoid such contamination of the literature, it is essential that toxicologists work with pure active ingredients, or in the case of formulated products, specify the formulation (which has to list any synergists, if there are). So I would still welcome a statement by the authors that the product used by them in this paper does not contain a synergist. For the future, it is advised to test pure deltamethrin and compare its toxicity with the same exposure from a formulated product.
Author Response

(The authors gave the same response as above.)
